# Theory and Application of Zero Trust Security: A Brief Survey

**DOI:** 10.3390/e25121595

**Published:** 2023-11-28

**Authors:** Hongzhaoning Kang, Gang Liu, Quan Wang, Lei Meng, Jing Liu

**Affiliations:** 1School of Computer Science and Technology, Xidian University, Xi’an 710071, China; kanghzn@stu.xidian.edu.cn (H.K.); qwang_xd@163.com (Q.W.); 23031212034@stu.xidian.edu.cn (L.M.); 2School of Computer Science and Engineering, Xi’an University of Technology, Xi’an 710048, China; liujing@xaut.edu.cn

**Keywords:** zero trust, network security, internet of things, cloud computing

## Abstract

As cross-border access becomes more frequent, traditional perimeter-based network security models can no longer cope with evolving security requirements. Zero trust is a novel paradigm for cybersecurity based on the core concept of “never trust, always verify”. It attempts to protect against security risks related to internal threats by eliminating the demarcations between the internal and external network of traditional network perimeters. Nevertheless, research on the theory and application of zero trust is still in its infancy, and more extensive research is necessary to facilitate a deeper understanding of the paradigm in academia and the industry. In this paper, trust in cybersecurity is discussed, following which the origin, concepts, and principles related to zero trust are elaborated on. The characteristics, strengths, and weaknesses of the existing research are analysed in the context of zero trust achievements and their technical applications in Cloud and IoT environments. Finally, to support the development and application of zero trust in the future, the concept and its current challenges are analysed.

## 1. Introduction

With the development of enterprises and economies, digitalization, data, and networks have become indispensable. To adapt to high-speed digitalization, data interaction and access have been boosted by increasing the number of terminal devices. Although the expansion of networks and frequent transmission of data has brought significant convenience, the sharply increasing internal threats within networks cannot be ignored. Team FireEye [1] noted that the proportion of external and internal threats changed from 94% and 6% in 2011 to 47% and 53% in 2021, respectively. The risks posed by internal threats and the costs of dealing with them have increased drastically, necessitating more in-depth research on network security and more concerted efforts to defend against internal threats. Zero trust was proposed to address this dilemma in traditional network security.

Traditional network security is based on the concept of a security perimeter, whereby the network is divided into two parts: an internal trusted network and an external untrusted network [2]. Based on this partition criterion, a well-structured defensive architecture treats the security of the network as an onion (see Figure 1), and each perimeter protects the area it covers. Northcutt et al. [2] defined a perimeter as the fortified boundary of the network, which may include border routers, firewalls, IDSs, IPSs, VPN devices, software architecture, DMZs, and screened subnets. However, this security perimeter is only a defence with a single direction and is powerless against attacks from within the network. Therefore, to protect against both internal and external threats, the security perimeter must be changed.

The concept of zero trust, “never trust, always verify”, was first proposed by John Kindervag in 2010 to address the issues caused by insider threats to the enterprise [3]. At its core is the idea of deperimeterization (limiting implicit trust based on network location) in recognition of the limitations of relying on single, static defences over a large network segment [4]. Based on the flaws of security perimeters facing internal threats and the risks arising from implicit trust, Kindervag further proposed three principles for zero trust security: (1) all sources must be verified and secured; (2) access control must be limited and strictly controlled; (3) all network traffic must be inspected and logged. These principles are the basis of zero trust. Integrating security environments and requirements with reality, zero trust achieves the goal of three aspects of security: application and device security, authentication and access control security, and network architecture security. As researchers extend the principle according to issues in actual scenarios, it is gradually becoming an emerging cybersecurity model.

Unlike other security paradigms, the application of zero trust has been developed in parallel with the study of its theory, especially in cyber enterprises. Google proposed BeyondCorp, a new zero-trust-based security method for its internal networks that eliminates privileged corporate networks. In the proposed method, all access to enterprise resources must be fully authenticated, authorized, and encrypted based upon the device state and user credentials [5,6,7]. By 2017, the method became fully implemented in the Google office network. It proved to be secure and made critical resources easily accessible when remote work became the norm during the COVID-19 outbreak. As zero trust gains wide attention, zero trust security is receiving greater scholarly attention, as more scholars attempt to address network security issues using abstract methods and architectures. Cloud Security Alliance (CSA) collaborated with a group of scholars to propose the Software-Defined Perimeter (SDP) under the premise of “need to know” [8]. By separating the control channel and data channel, the resource host, which is exposed by the original port, is black-boxed from the communication layer. Thus, authentication and authorization are required, in order to know or access the resources in each transaction. The National Institute of Standards and Technology (NIST) integrated the research on zero trust and proposed the zero trust architecture (ZTA) as a basic security paradigm [9]. The NIST proposed logical components that separate the control plane and data plane from the network layer, which is suitable for organizations with a highly mobile workforce. Thus, the ZTA can provide effective support for the implementation of zero trust in distributed mobile scenarios. Consequently, zero trust can be applied in centralized or distributed environments such as Cloud and IoT. In addition, scholars have analysed and summarized the security mechanisms and critical technologies in zero trust from different perspectives [10], further promoting its academic development and application of zero trust.

Referring to the zero trust literature search method in [11], the zero trust literature from 2014 to 2022 in the Web of Science Core Collection database are searched and sorted out, as shown in Figure 2. It can be observed that a large amount of research has appeared in the field of zero trust since 2019. The emergence of this phenomenon shows that zero trust, as an emerging security concept, is gradually accepted by researchers and that it has a large number of directions to be explored. To help researchers understand the basic knowledge of zero trust more comprehensively, representative literatures on zero trust theory and application are selected among the 1027 retrieved documents for analysis. The literature on zero trust concepts, theory, achievements, and applications are systematically reviewed in this paper. On this basis, the difference between zero trust security and traditional perimeter security is innovatively compared from the perspective of trust itself, and the concept, characteristics, and basic principles of trust in zero trust is proposed. In addition, in view of the current research status and development trends of zero trust, the current challenges of zero trust and the directions that can be explored in the future are also organized from the perspective of the trust in zero trust.

The main contributions of this paper are:Innovatively understand zero trust from the perspective of trust in network security, and discover the existence of trust in zero trust through representative literature on zero trust theory and application.Proposes the concept and characteristics of trust in zero trust, and provides the basic principles it should have. On top of this, the research trends of zero trust in different scenarios in the future are discussed.

The structure of this paper is as follows. Firstly, the origin, concepts, and deployment of zero trust are introduced. In the Section 2, the trust theory, as well as the definition and principles of zero trust, are introduced and the different successful implementations of zero trust, such as SDP, ZTA, and BeyondCorp, are presented. The applications of zero trust in the Cloud and IoT are briefly introduced in the Section 3. In the Section 4, we analysed the concept of trust in zero trust and extend it through a definition and three principles. The Section 5 presents the challenges of zero trust and the future research direction. The Section 6 is the summary of the paper.

## 2. Conceptual Background

Despite being introduced ten years ago, the concept of zero trust is still in its early stages as an emerging network security concept. This section attempts to study zero trust from the perspective of trust and explores the representative literature on trust in cybersecurity, the concept and principles of zero trust, and its achievements. Finally, a table is provided to showcase the relevant literature.

### 2.1. Trust in Cybersecurity

The concept of trust predates cybersecurity and has been discussed and analysed by sociologists for many decades. Rousseau et al. [12] define trust as “a psychological state comprising the intention to accept vulnerability based upon positive expectations of the intentions or behaviour of another”.

Although this generally accepted definition captures an inter-disciplinary perspective, it does not fully capture the dynamics and varied subtleties of trust. Thus, there is no universally accepted scholarly definition of trust. The implications and classification of trust in cybersecurity has always been determined by the context. According to Govindan et al. [13], trust can be reflected in reliability, utility, availability, reputation, risk, confidence, quality of services, and other concepts. However, none of these concepts accurately capture trust. For example, although some security professionals regard trust as a security metric or an evaluation methodology [14], others consider it as a relationship between entities [15]. The focus and security requirements of different scenarios impact trust, imbuing it with ambiguity. Thus, scholarly attention has shifted from the definition of trust to the classification.

With complexity and ambiguity, trust is classified based on the context. In safety culture, Burns et al. [16] divided trust into explicit and implicit trust. Explicit trust is derived from the clear standard that people use, relevant information obtained, and existing laws and regulations to objectively and fairly judge the credibility of others. This improves safety by sacrificing practicality. Implicit trust, on the other hand, derives from people’s subjective perception of the trustworthiness of others based on emotions and experiences. This sacrifices a certain amount of security for improved practicality. According to Dunning et al. [17], trust is constructed based on people’s ethical norms, rather than established societal norms, whose influence dwindles, as mutual trust between two parties deepens. From the perspective of the weight change of explicit and implicit trust, with the deepening of mutual trust between two parties, the objective explicit trust determined by societal norms will gradually be weaker than the subjective implicit trust accumulated through multiple interactions; the increase in the influence of implicit trust will further promote the conclusion of subsequent transactions. Thus, trust comprises objective explicit trust and subjective implicit trust, and the weight of the two directly affects the safety and practicality of trust in actual use.

In cybersecurity, there are more refined trust classifications. Govindan et al. [13] classified trust as a risk factor, belief, subjective probability, or transitivity. For mobile ad hoc networks, trust is more of a subjective assessment. The reliability and accuracy of the received information should be assessed in a given context. Trust can reflect the belief, confidence, or expectations of the target node’s future activity/behaviour and the mutual relationship between the nodes that behave in a trustworthy manner with each other. Pearson et al. [18] surmised that both persistent trust and dynamic trust are required in cloud computing. The major difference between persistent and dynamic trust is the length of the trust life cycle. Persistent trust is derived from long-term underlying properties or infrastructure; dynamic trust, on the other hand, exists briefly in specific states, contexts, or for single information. Thus, the reliability of the former is more dependent on the long-term existing mechanisms of society or industry, and the latter is closer to trusted computing availed by modern computer technology. However, these definitions and classifications of trust always rely on the traditional perimeter to divide trusted and untrusted zones. The gradual disappearance of the traditional perimeter poses a challenge that impels a new security solution.

### 2.2. What Is Zero Trust

Prior to zero trust, the default assumption among security professionals was that all the data and transactions inside the perimeter were always trusted [19]. However, risks, such as penetration attacks, malicious insiders, and loss of data, degrade trust. Trust for malicious insiders is degraded by the system only after the target resource is obtained. Owing to this, critical resources are always under threat, even within the secure perimeter. Furthermore, in current authentication and authorization methods, malicious insiders retain permissions until the trust is re-evaluated. To address the flaws of the traditional perimeter, zero trust addresses insider threats in an internal network using deperimeterization [3]. It describes a transition that reduces or even eliminates the perimeter and secures the system using a continuous approach that verifies each device, user, transaction, and even data flow, during the entire access process. Thus, it assumes the stance of “never trust, always verify”. However, there is no specific and universal definition of trust. Most scholars regard it as a cybersecurity paradigm related to identity authorization, fine-grained access control, and secure communication that focuses on combining existing technologies. Others consider zero trust a cybersecurity paradigm focused on resource protection and have attempted to summarize it as several abstract definitions and architectures that eliminate implicit trust from a more fundamental perspective [9]. In general, zero trust is concerned with the use of technology for the comprehensive, accurate, and real-time control of security systems.

Owing to the growing sophistication of AI, it is being introducing into security systems. For zero trust, AI, as a human analogue, is becoming one-sided, in terms of assessing security using technical metrics alone. Tidjon et al. [20] attempted to understand the factors influencing the trustworthiness of an AI system. By compiling and summarizing the literature, transparency was found to be the most adopted principle. Theoretically, trustworthiness can be judged based on the 12 attributes of transparency: privacy, fairness, security, safety, responsibility, accountability, explainability, well-being, human rights, inclusiveness, and sustainability. This finding illustrates the need for researchers on zero trust to introduce human-related factors to further enhance security, in addition to the constant enhancement of technology. Regardless of how advanced AI technology becomes, it must ultimately be used only when humans trust it. This, in turn, is one reason why the definition of zero trust is controversial.

Therefore, although there is no unified definition of zero trust, existing research has provided an understanding of its core principle. Based on the works of Kindervag [21], ACT-ICA [19], NIST ZTA [9], the National Security Agency [22], and J. Garbis et al. [23], we reviewed the principles of zero trust and made the following deductions:**Separation of trust from location**. This principle is one of the basic premises for achieving zero trust. The biggest difference between zero trust and traditional security perimeters is whether the location determines the trust in the access behaviour. Zero trust dispels the credibility of the internal trusted network set by the traditional security perimeter based on the resource location, and is premised on the belief that location can no longer fully guarantee trust in the current network environment. Furthermore, the network security situation, such as the long-term hostility of the network and existence of internal and external threats, makes the trust gained by the location unable to guarantee the security of critical resources in the network. The separation of trust from the location that is the core of zero trust can invalidate the trust determined by location, thereby reducing the scope of the influence of implicit trust in internal trusted networks, and ultimately achieving the goal of resisting threats from internal and external networks simultaneously. However, it should be noted that that trust is not solely determined by location, as zero trust does not completely negate the influence of the location on trust judgment; rather, it is simply one of other collectible elements as an equal condition for judging trust.**The principle of least privilege**. The formulation of least-privilege policies is essential to achieve frequent and fine-grained authentication and authorisation. All requested permissions must be restricted to a specific entity under access and only given the minimum permissions for the current operation. This is similar to the principle of the least privilege in the access control. It is necessary to enumerate all possible access conditions and avoid conflicts among policies by comparing a series of elements related to access such as subjects, resources, and context. At the same time, the principle of the least privilege can also reduce the scale of risks caused by the abuse of power and minimize the scope of threats. Additionally, dynamic security policies must be used to maintain the necessary flexibility in dynamic contexts. Therefore, the scale of zero trust security policies is often determined based on the complexity of the deployment scenario.**All data and services as resources**. Zero trust expands resource coverage and protects critical resources from damage. Logically, the access can be regarded as the operation of the subject in a specific environment, aiming to protect against existing attacks. However, once services involving data flow and computation are damaged, access security will also be affected, and protection against attacks cannot resist unknown attacks. Therefore, all data and services included in the zero trust access process are regarded as resources that are as important as the object or device being accessed, and critical resources are specifically protected.**Continuous monitoring and evaluation**. No entity is inherently trustworthy; thus, all entities should be monitored. The monitoring proposed here does not monitor only specific threat behaviors or characteristics as before, it monitors all states of all entities (data flow, devices, services, files) related to the access. A robust continuous monitoring system can collect environmental information as much as possible and provides reliable data for safety assessment. As the observable information increases, the credibility of the security analysis results obtained by the assessment system would increase, thereby reducing the probability of threats caused by the trust.

Zero trust is a cybersecurity paradigm, holistic model, systematic approach, and set of guiding principles. Although it can be interpreted in many ways in different scenarios, its principles are constant. Researchers have always realized zero trust in applications and devices, authentication and access control, and network architecture. From these perspectives, zero trust and its deployment can be analysed intuitively.

### 2.3. Zero Trust Achievement

Zero trust is a cybersecurity paradigm wherein no user, transaction, or network traffic is trusted, unless verified [24]. Based on these principles, it can be guaranteed mainly through four aspects: authentication, access control, continuous monitoring, and evaluation [25]. These components are closely combined to realize the final security system. Currently, there are a few zero trust approaches in academic and industrial research, and these achievements can help scholars understand how to achieve zero trust.

To address the security risks of the distributed denial of service attacks and sniffing attacks owing to unauthorised access to infrastructure, CSA follows a “need-to-know” model, and proposes SDP as a security model/framework that dynamically protects modern networks [8,26,27]. Further, the CSA proposes that the premise of a traditional enterprise network architecture is to create an internal network demarcated from the outside world by a fixed perimeter. The architecture consists of a series of firewall functions that block external users and allow internal users to exit. However, the traditional fixed-perimeter model is rapidly becoming obsolete. Bring-your-own-devices and phishing attacks have resulted in untrusted access, and the location of the perimeter has been changed by software-as-a-service and infrastructure-as-a-service. This is similar to the internal threat issues that zero trust attempts to fix, and the researchers regard SDP as an approach to zero trust. SDP, which differs from a traditional system, affords visibility to everyone but allows connectivity on a “need-to-know” basis by adding several point control systems. In a real system, SDPs replace physical appliances with logical components that operate under the control of an application owner. This implies that it transfers the responsibility of granting trust to the requester from the application designer and then to the resource owner.

The architecture of the SDP consists of two components: SDP hosts and SDP controllers. SDP hosts can initiate or accept connections. These actions are managed through an interaction with the SDP controller via a secure control channel (see Figure 3). Thus, the control and data planes are separated to realise a completely scalable system. In this architecture, the SDP controller undertakes the task of performing authentication before access and controls the opening and closing of the data channels between hosts. Thus, SDP can effectively defend against remote hypervisor attacks, denial of service attacks, virtual machine hopping, and port scanning [28]. Five separate security layers compose and support the SDP architecture: single packet authentication (SPA), mutual transport layer security (mTLS), device validation (DV), dynamic firewalls, and application binding (AppB). Among them, the SPA is the basic component for maintaining secure authentication and valid traffic before the connection between the parties of the transaction. The SPA requires that the first packet be cryptographically sent from the initiating host to the SDP controller, where the host’s authorisation is verified before granting it access. Subsequently, the SPA is sent by the host to the gateway to help it determine the authorised host’s traffic and reject all other traffic. With these components and technologies, the SDP can effectively defend against attacks from insiders and protect critical resources.

The zero trust architecture (ZTA) was first proposed by NIST in 2020 [9]. Compared to the SDP, the ZTA is a systematic security architecture that contains technologies such as SDP, access control, and multi-factor authentication. This supports the logical component of ZTA (see Figure 4) that separates the control and data planes. The policy decision point (PDP) is the core of the control plane and is responsible for authentication and authorisation. It grants access based on trust that satisfies the security policies of the system. The policy enforcement point (PEP) is the core of the data plane responsible for transporting the trust from the PDP to the current access of the subject and permits the connection between the subject and enterprise resources. By demarcating the control and data planes, the ZTA can effectively integrate the related technologies and modularly add them into the architecture.

Syed et al. [29] analysed and collated the basic principles and related technologies of the ZTA. They also derived seven directions for zero trust: lightweight and scalable continuous authentication techniques, fine-grained context-based access, data encryption under resource constraints, microsegmentation techniques to cope with single points of failure, threat-aware systems that integrate heterogeneous data sources and monitoring logs, reliable automated trust assessment knowledge systems, and application-level access control enforcement procedures. As can be observed, the implementation of zero trust relies on the combination of multiple technologies.

To accelerate the deployment and implementation of zero trust, the NIST categorises the mainstream technologies for it into three topics from a technical perspective, namely SDP, identity and access management (IAM), and micro-segmentation (MSG), which together are known as “SIM”. IAM acts as a web service to ensure the secure access to resources by controlling the authentication and authorisation. As an application technology predating ZTA, IAM systems have already been offered by many organisations such as AWS [30], SailPoint [31], IBM [32], Oracle [33], RSA [34], and Core Security [35]. MSG, on the other hand, is a network security technology that is concerned with the isolation of horizontal traffic in the network [36]. By dividing all the services within a data centre into several tiny network nodes according to specific rules, MSG can enforce the access control on these nodes through dynamic policies, thereby achieving logical segmentation. From the perspective of zero trust deployment, SDP provides the technical foundation at the network data level, IAM provides a viable management method for continuous authentication and fine-grained access control, and MSG delineates the logical areas of business data. With these technologies as its foundation, zero trust has progressed rapidly in a short period.

BeyondCorp, proposed by Google for its enterprise security, is also recognised as a valid solution for zero trust. Based on the assumption that an internal network is fraught with as much danger as the public Internet, BeyondCorp is a new security model that dispenses with a privileged corporate network [5,6,7]. It requires that all access to enterprise resources be fully authenticated, authorised, and encrypted based on the device state and user credentials. According to the fine-grained access to different parts of enterprise resources, the user experiences of local and remote access to enterprise resources are identical. The major components of BeyondCorp are shown in Figure 5. To remove trust from the network, BeyondCorp defines and deploys an unprivileged network that closely resembles an external network, which is still within a private address space controlled by Google itself. Furthermore, BeyondCorp uses a strictly managed access control list (ACL) with the information of all the client devices related to Google. The ACL controls access between different parts of Google’s network using the access control engine.

As a critical technology in networks, the security of Docker deployments is noteworthy. Leahy et al. [37] investigated the security state of Docker containers deployed by default on Linux from a containerisation perspective and proposed a zero trust container architecture (ZTCA). The ZTCA builds on the strategic ideas and principles of the ZTA and successfully demonstrates that the zero trust principle can censor and secure a wide range of Docker use cases to ensure the security of Docker deployment. This demonstrates that the relevant security principles at the core of zero trust can not only theoretically secure systems, but also provide security enhancements to existing technologies. However, the proposed framework has the same limitations as the other security frameworks. It is complex to deploy and requires a high level of security personnel, which are common problems faced by existing zero trust frameworks. Therefore, simplifying the design process, while ensuring security, is critical for future zero trust research.

Taking a comprehensive look at various zero trust architectures, it can be observed that zero trust architecture is a holistic solution that encompasses the entire life cycle of a network. It covers aspects such as identity authentication, access control, data protection, network security, application security, and threat monitoring. By integrating these technologies into the architecture, zero trust achieves its security objectives. From a trust perspective, it is the trust established through these aspects that ensures the overall security of the zero trust architecture.

In the context of identity authentication, trust is derived from the verification of the legitimacy of the requesting entity, and it serves as a fundamental consideration when establishing a zero trust architecture. Whether it is traditional user authentication mechanisms or more context-aware and continuous identity authentication methods, as well as device authentication mechanisms that focus on digital identity, these identity authentication technologies hold the same significance within the zero trust architecture. Security practitioners of zero trust need to selectively apply appropriate technologies based on specific scenarios and business requirements.

In the realm of access control, trust originates from the effective granting and restriction of permissions to entities during access, which forms the fundamental guarantee for the principle of least privilege in zero trust. Identity-based, role-based, attribute-based, intent-based, and risk-based access control mechanisms, which provide assistance in access granularity, permission granting, and policy management, are all crucial considerations for the design and implementation of a zero trust architecture. These mechanisms should be taken into account by zero trust security researchers. Regarding data protection, trust relies on the reliability of encryption algorithms, and the implementation of a zero trust architecture necessitates the selection of suitable solutions based on specific security requirements, data formats, and computational resources of the given scenario.

In terms of network security, trust is built upon the effectiveness of network segmentation, which is an essential aspect that needs to be determined prior to deploying a zero trust architecture. Different segmentation strategies will influence the choice of specific technologies in other areas, ultimately impacting the flexibility of the zero trust architecture.

Application security and threat monitoring, on the other hand, further mitigate uncertainties introduced by human factors within a zero trust architecture. By promptly detecting and responding to abnormal events, these practices help reduce security risks.

All of these zero trust architectures seek to separate data and control to a great extent and achieve the security requirements of zero trust through the joint management of different control components. However, it is important to note that the ultimate goal of zero trust is to protect against insider threats that existing security models cannot protect against, which requires it to be deployed in a realistic manner to validate theoretical and methodological feasibility and reliability. In addition, research on insider threats is an area on which zero trust researchers need to focus. Only a comprehensive understanding of insider threats can render zero trust research results usable and feasible for solving practical problems.

### 2.4. Overview of the Literature

Table 1 summarizes the basic information about zero trust theory and architecture-related literature, and gives the author’s views on zero trust.

## 3. Application

The IT landscape is empowered by a connected world that is more susceptible to malicious activity owing to its connectedness, user diversity, wealth of devices, and globally distributed applications and services. The complexity of the current and emerging Cloud and IoT has exposed the lack of effectiveness of traditional security perimeters. These issues can be addressed separately by fine-grained access control, continuous authentication, log audition, and network microsegmentation. As a cybersecurity paradigm combining these technologies, zero trust can address the issues of the traditional security perimeter and can be applied to Cloud and IoT. This section introduces partial zero trust solutions for Cloud and IoT scenarios and analyses the focus areas in these two contexts. Additionally, the last section provides an overview table of the relevant literature.

### 3.1. Application of Zero Trust in the Cloud Environment

The benefits of clouds include virtual computing technology, a powerful storage capacity, and good system scalability. This makes them more familiar and valuable to enterprises and scholars. With the increasing scale and complexity of in-cloud environments in recent years, insider attacks against clouds have increased significantly. However, most clouds still adopt a traditional perimeter defence, which leaves them without effective defences against insider threats, especially data loss, theft, and destruction caused by lateral attacks in the cloud. The proposal of zero trust provides a new solution to these issues, and scholars have introduced zero trust in cloud environments with appropriate adaptations to the original technology [38,39,40,41].

The traditional security perimeter simply divides a network into an internal trusted network and an external untrusted network. This prevents the perimeter from defending against insider threats in the internal network and creates risks for the critical resources. For this purpose, Huang et al. [38] proposed a framework for analysing trust relations in the cloud. The trust mechanism comprises cloud service trust, service provider trust, cloud broker trust, cloud auditor trust, and societal trust. This fine-grained segmentation enables trust to be applied on a smaller scale.

Considering that the best security practices adopting network segmentation in traditional data centres are not well suited to cloud computing environments, C. DeCusatis et al. [39] proposed a zero trust cloud network segmentation method achieved by transport access control (TAC) and first packet authentication. They combined both approaches into a single unified defence to realize zero trust in the cloud environment. Each network session must be independently authenticated at the transport layer before any access to the network or protected servers is granted. In addition, explicit trust is established by generating a network identity token attached to the first packet of a TCP connection before the data traffic of sessions between the client and server (see Figure 6). However, this approach to security is predicated on modifications to the transport layer protocol, which imposes a significant overhead on cloud service providers that have already deployed operations.

Owing to the emerging microservices of the cloud, attacks can propagate laterally within the data centre by exploiting cross-service dependencies. Thus, Zaheer et al. [40] proposed shifting the perimeter from network endpoints to workflows. They assumed that the infrastructure provider and information from the trusted infrastructure were trustworthy. Based on this assumption, they proposed an extended Berkeley Packet Filter that could track the context of a microservice workflow. Thus, the perimeter of the workflows could be changed by detecting the workflow data and the context of the provider. This method implements zero trust for applications and enhances security in a microservice cloud environment.

Zolotukhin et al. [42] used the defence idea of the SDP to deploy deep learning components on top of the software-defined networking (SDN) and network function virtualization (NFV) controllers in the SDN and NFV technologies, thus enabling the real-time detection of network states and dynamic adjustment of security policies. However, the practicality of the approach was not effectively proven, as the researchers used artificially generated network traffic data. By empowering network security management components with intelligence through reinforcement learning algorithms, the components could no longer be limited to static and unchanging security policies, which is in line with the reality of dynamically changing network environments and security requirements.

Comparing the zero trust solutions in cloud scenarios reveals that the focus is primarily on addressing the interactions between large-scale data and services in the cloud, efficient network segmentation within the cloud, and the design of effective trust evaluation and auditing mechanisms. These three requirements highlight the key considerations for zero trust in cloud scenarios: ensuring secure communication for large-scale interactions, enabling dynamic network construction, and establishing explicit trust relationships. From a trust perspective, the interactions between services in the cloud outweigh those between individuals and services. This implies that establishing and maintaining trust relationships between services is of paramount importance for zero trust in the cloud. Additionally, the extensive interactions in the cloud generate a significant amount of recordable data. Leveraging information theory concepts such as information entropy, source coding, and channel coding can provide additional insights into the state of the cloud, enabling more comprehensive network intelligence and enhancing communication reliability and transmission rates.

In addition to the security components in the cloud, humans are also a source of risk with which zero trust is concerned. Sarkar et al. [43] surveyed several implementations of a zero-trust-based cloud-network model. Different methods and applications for authenticating and authorising key services used in a trust-based cloud network were examined; it was found that there were various problems in moving from existing system architectures to a ZTA for deploying zero trust in a cloud environment. Among these, the most obvious impediment was humans. Zero trust focuses on more granular data than other security architectures and, therefore, may also carry the risk of compromising privacy. Cloud is the dominant storage and management environment in business today, and zero trust cannot be accepted by the general public, regardless of how secure it is, if a user or organisation discovers that the privacy of their data on the cloud has been compromised. Therefore, to deploy zero trust in the cloud, the architecture and approach must be designed for data privacy.

Although these zero trust solutions can provide a better protection for resources in the cloud, there are still some shortcomings. First, most of the existing zero trust solutions adopt a three-element architecture in which the control and data planes, that is, the subject, controller, and object, are separated (see Figure 7). The controller serves as the central node for the authentication and access control, and its ports are completely open to the entire network. Once the controller has a single point of failure or is attacked, the execution efficiency of the entire zero trust solution is significantly reduced, and its security is affected. If a distributed controller is used, the consistency of information between the controllers becomes a key feature that affects the accuracy of the execution result. Although the strong consistency technology of the blockchain can be used to ensure the synchronisation of information between the various controllers, the performance and time overhead caused by the blockchain will become an important factor affecting the availability of the program. Therefore, zero trust solutions in the cloud environment must be developed for specific needs at the expense of certain indicators, in exchange for the improvement of other indicators.

### 3.2. Application of Zero Trust in the IoT Environment

The IoT is an Internet system that integrates various sensors and objects to communicate with one another without human intervention. Security and privacy issues have become increasingly evident with the large-scale deployment of IoT [44]. Unknown devices and traffic can exacerbate the spread of vulnerabilities between interconnected devices in sensitive locations with access to potentially harmful actuation capabilities [45]. With the rise of AI and machine learning technology, IoT technology has gradually improved, becoming one of the development trends in society. However, the large number of devices moving in and out of the IoT makes it difficult to deploy a fixed perimeter, and device management has become increasingly complicated. With the deepening of research, scholars believe that zero trust, with the requirement that all devices must be verified, whether they are inside or outside, can address these issues, thus ensuring the security of the entire IoT.

With its advantages of distributed databases, smart contracts, consensus, and immutability, blockchain has become a popular technology for achieving zero trust in IoT. Samaniego et al. [46] proposed Amatista, a blockchain-based middleware, and applied it to achieve hierarchical zero trust management in IoT. This is a novel zero trust hierarchical mining process that allows different levels of trust, to validate infrastructure and transactions. The shift from a centralised to a distributed approach for trust management and mining enables the deployment of zero trust in the IoT environment. Dhar et al. [47] proposed a peer-to-peer blockchain network framework that operates in parallel with a zero-trust-based security architecture. Components, such as the segmentation gateway, microcore, perimeter (MCAP), and management server are connected to the blockchain. It addresses the security concerns of risk-based MCAPs and cryptographically secures storage and transmission. Zhao et al. [48] introduced blockchain as an authentication scheme for IoT devices, making it possible to switch smart devices from an untrusted state to a trusted state. They all leverage the distribution and immutability of blockchain to deploy zero trust in the IoT.

Researchers have further explored the relationship between zero trust and blockchain through studies from both theoretical and application perspectives. Alevizos et al. [49] conducted a comparative analysis of the traditional perimeter-based model and zero trust model, and explored its potential use for endpoints based on blockchain foundations. Their study showed the capability of endpoint integrity testing, demonstrating that the blockchain technology is indeed capable of supporting endpoint authentication for ZTA. However, it should be noted that the biggest impediment to the application of blockchain technology to zero trust is the significant overhead associated with the technology itself. The fine-grained authentication and access control necessary for zero trust affect the usability of blockchains in real-world scenarios. From this perspective, to apply blockchain technology to zero trust, research might be predicated on two ideas: one is to simplify the consensus algorithm and data structure in the blockchain to accommodate the constrained endpoint resources, and the other is to use multiple endpoint clusters to increase the computing power of blockchain nodes.

To use blockchain technology to achieve zero trust for IoT, researchers are also attempting to optimize existing ZTAs and zero trust technologies. Palmo et al. [50] found that ensuring the reliability of IoT itself is critical when embedding IoT devices into SDPs. In this regard, they analysed the federation evaluation method of the IoT gateway, federation evaluation method of identity provider (IdP) and federation evaluation method of certification authority. From a qualitative evaluation perspective, it was determined that the IdP federal evaluation method had the least overhead and was the easiest to administer and install. However, for the application of zero trust in IoT, the existing research lacks validation in simulated or actual IoT environments. A purely qualitative evaluation can only prove the feasibility of the method from a theoretical perspective, not its effectiveness and usability in reality.

How zero trust can be deployed in the IoT environment under 5G is also a current research priority. Valero et al. [51] proposed a new security and trust framework for 5G multidomain scenarios and validated zero trust principles in distributed multistakeholder environments. The security and trust levels of multi-stakeholder 5G networks are improved through trust and intra-domain and inter-domain modules. This hierarchical approach to security rules allows individual stakeholders in a 5G network to focus more on their own security-related matters, thus avoiding unnecessary overheads. However, the dynamic nature of stakeholders cannot be ignored. How this security and trust framework automatically adjusts when stakeholders change is the next critical issue to consider. At the same time, the trustworthiness of the AI approach used in the security and trust framework is also a difficult issue in zero trust research.

Li et al. [52] also provided an outlook on the security of future industrial IoT, resulting in a blockchain-based zero trust architecture for future IoT. To cope with the complexity and performance requirements of 5G-IoT systems, they proposed specific frameworks to achieve the zero trust authentication of devices/users/applications. However, 5G-IoT poses significantly more security issues than traditional IoT, particularly heterogeneity and interoperability. Zero trust research under 5G-IoT could mitigate the lightweight requirements and focus on how to provide a set of architectures with compatibility, scalability, and different granularity of the access control.

Some scholars believe that zero trust can address the issue of power IoT security. The power IoT has a massive terminal access and facilitates efficient information sharing, while addressing the problem of increasingly blurred grid boundaries. Chen et al. [53] proposed the use of blockchain to enhance the security of data interaction and achieve a high-level protection of data circulation in all the links of the power network. They used blockchain as a computing paradigm and collaboration model to establish trust at a low cost in an untrusted competitive environment. Based on the ZTA, Xiaojian et al. [54] proposed a power IoT security protection architecture for network boundary and channel security protection, business application security function design, and mobile terminal software security protection. They used a central policy library to manage dynamic access-control authorisation strategies. In addition, they reduced the granularity of access to a single operation on a single device. These studies proved that zero-trust-related technologies can effectively address security issues in IoT environments.

In contrast to the close relationship between zero trust solutions and services in cloud scenarios, zero trust solutions in the IoT focus more on integrating with blockchain technology and addressing the requirements of different real-world scenarios. In terms of blockchain integration, the emphasis of zero trust solutions lies in leveraging the distributed nature of blockchain to achieve multi-level trust management and designing device authentication schemes based on information consistency. However, the incorporation of zero trust’s fine-grained identity and access control introduces significant overhead, which also affects its integration with blockchain. In addressing the requirements of different real-world scenarios, there are diverse demands on data interaction, communication methods, data management, and scalability in zero trust solutions due to the dynamic nature of 5G networks, the complexity of industrial IoT, and the security and reliability of smart grids. While existing zero trust solutions in IoT and blockchain are already capable of achieving some functionalities, the inherent limitations of IoT still pose challenges to the design and implementation of zero trust solutions.

To further deploy zero trust in IoT environments, future researchers should focus on the following three issues. The first is the deployment of zero trust technologies under resource constraints. Current zero trust technologies tend to prioritise security over lightweighting, but most IoT devices cannot afford the computational overhead (such as deep learning) that significantly exceeds the computation required by their own business. Therefore, lightweight deployment is the first issue to be addressed in zero trust deployments under IoT. The next issue is how to cope with the impact of changing device dynamics on zero trust deployments. In reality, the movement of IoT devices can have an impact on factors such as transmission rate and network topology, thus changing the state of zero trust deployment. Finally, the interoperability of heterogeneous devices is an issue. There are many different types of IoT devices with differences in device models, message formats, and transmission methods. One of the key principles of zero trust security is data monitoring and management. This heterogeneity can hinder the deployment of zero trust in the IoT. An architecture or protocol that supports the interconnection of heterogeneous devices can significantly facilitate the deployment of zero trust in IoT.

### 3.3. Overview of the Literature

Table 2 summarizes the basic information about zero trust literature under cloud and IoT, and gives the author’s views on zero trust.

## 4. Analysis

Zero trust is a cybersecurity paradigm first proposed by Forrester for insider threat issues experienced by enterprises and was subsequently studied by scholars. It has proven to be an efficient way to address the issues caused by insider threats in internal trusted networks and has been deployed by Google, Microsoft, and Gartner, among others. Thus far, researchers of zero trust have focused on architecture and framework, neglecting trust. However, the trust in zero trust is also concerning, and it is the key to ensuring the safety from untrusted to trusted [19]. In different scenarios, the trust in zero trust has various features and sources, and the security method should change as the context-based trust requirement changes. Based on the trust theory in sociology and cybersecurity, we analysed the concept of trust in zero trust and the core principles.

### 4.1. Trust in Zero Trust

Unlike the literal meaning of zero trust, the “zero” here does not imply an absolute absence of trust; rather, it indicates zero inherent or implicit trust. It has been proven in sociology that with the deepening of mutual trust between two parties, the objective in explicit trust determined by social norms will gradually be weaker than the subjective in implicit trust accumulated through multiple interactions, and the increase in the influence of implicit trust will further promote the conclusion of subsequent transactions [17]. Thus, the target of zero trust is not to eliminate all trust but to eliminate implicit trust and enhance the authentication security of explicit trust.

Zero trust is more concerned with the security of the resource itself. The resource owner should be regarded as the trust initiator and risk bearer, and the code of conduct is an important basis for evaluating trust in every transaction, based on the “never trust, always verify” stance. To summarise, trust in zero trust is a type of minimum permission to facilitate the achievement of the transaction and satisfy safety standards. It is a risky decision made by the resource owner based on the intersection of the codes of conduct that both parties follow.

Safety standards are a series of behavioural norms formulated by safety personnel according to the context. Minimum permission means that the granted trust needs to satisfy the principle of least permission for access control. The intersection of the codes of conduct is the same part of the behavioural norms that requesters have shown in the current network and are the behavioural norms required by resource owners. Zero trust requires that each granted trust be authenticated, and authentication is achieved by referring to a mutually agreed approach by the transaction parties. The source of this mutually agreed approach is the essence of trust, which is generated by the intersection of the codes of conduct that both parties follow. Thus, the essence of trust is the security instance produced by the intersection of the code of conduct, and the components of trust should be changed from explicit to implicit trust.

In zero trust, implicit trust cannot guarantee security and validity, and objective instances of trust evaluation in the generative process of explicit trust may not be trusted. Thus, the trust composition of the original explicit and implicit trust cannot satisfy the requirement of zero trust. Therefore, there must be a more basic form of trust in zero trust to maintain security and validity when explicit and implicit trust fail simultaneously. We consider that the trust in zero trust has three parts: explicit trust, implicit trust, and trustbase (see Figure 8).

In zero trust, explicit and implicit trust are still considered an external manifestation of trust in the network. However, zero trust disapproves the security and validity of implicit trust and the minimum need for practicality in actual use. Implicit trust in zero trust can only be given in the smallest access granularity, which reduces the proportion of implicit trust. Therefore, the security of trust in zero trust depends on the agreement reached by the two parties before the interaction. Explicit trust, whereby the trust relationship between the two parties is maintained through compliance with the provisions of the agreement, has become the most frequent form of trust in zero trust. This also confirms why the current research on zero trust focuses on frequent and fine-grained authentication and access control. In addition, a component called trustbase is added as the basis of explicit and implicit trust. It comes from the above-mentioned concept of ‘the intersection of the code of conduct that both parties follow’, and it appears as a form of trust when both explicit and implicit trust fail.

In this regard, the trustbase is the basis for building trust, and it is a time-sensitive and non-verified security instance produced by the code of conduct that both parties follow in the current transaction. When explicit and implicit trust do not exist or fail simultaneously, the trustbase will promote the achievement of the transaction and maintain the minimum security requirements of the resource owner. This requires trust evaluation, a security characteristic of explicit trust. Maintaining the minimum security requirements of the resource owner promotes the conclusion of the transaction, which has the practical value of an implicit trust. Therefore, a trustbase can be used as an explicit trust framework to provide security for the trust in zero trust, and it can also be used as an implicit trust framework to provide practicability for trust, while also satisfying the security requirements specified by zero trust. Based on this inference, we can conclude that a trustbase can be used as the basis for building trust in zero trust and maintaining a trust-based security standard system in cybersecurity.

### 4.2. The Principles of Trust in Zero Trust

With the deeper research on zero trust, the principles are also expanding. Although the scenarios of each study are not the same, most follow similar principles. Thus, the trust in zero trust also has similar principles. We analysed the current work on zero trust and identified three principles for trust in zero trust.
**Trust should be context-based**. This principle is derived from the requirements for dynamic access control and continuous evaluation. In the existing deployment of zero trust, security professionals can realize authentication and authorization through fine-grained dynamic access control policies. These policies must address the security requirements of the system for access in different contexts and grant trust to the access that is met. This means that trust in zero trust is not persistent but dynamic, and should be context-based.**Trust should be based on the minimum security requirements of resource owners**. Unlike the traditional perimeter, zero trust defaults on all transactions, and data are untrusted until verified. Additionally, the principle of the least privilege allows trust to be granted to the finest-grained information carrier in the interaction such as a single transaction or packet. However, once the critical resource is leaked and destroyed, the resource owner will be exposed to great security risks. That makes the risk of leakage often borne by the resource owners. Thus, in zero trust, the minimum security requirements of resource owners must be satisfied before trust can be granted.**Trust should be hierarchical**. In realistic scenarios, different trust evaluation criteria may produce opposite results for the same matter, thus creating conflict. A hierarchy of trust should be established to ensure consistent results in the case of conflict. In zero trust, the hierarchy of trust is that trust has different priorities in different contexts. Owing to the complexity of the context of cybersecurity, there could be multiple trusts in zero trust. If there is no hierarchical division, the credibility assessment of current transactions based on different forms of trust will lead to inconsistencies in the assessment results, which will render the transaction unsuitable for normal processing. Therefore, to solve the problem of the difference among the trust evaluation results of different forms of trusts caused by the continuous change in context, there should be a clear classification of trusts according to context-based criteria.

## 5. Feature Research Trends

With the escalating internal threats and diminishing trust efficacy in the cybersecurity landscape, security models have become more rational and stringent, particularly in the context of zero trust. Embracing the “never trust, always verify” principle, zero trust necessitates authentication and authorization for each transaction during the access process between the requester and resource owner. Current research endeavors primarily focus on achieving deperimeterization and establishing defenses against internal threats. Researchers have proposed a range of architectures and implementations to address data interconnections in the context of zero trust, which have garnered widespread acceptance and adoption in numerous enterprises. In order to provide future research directions for scholars in related disciplines, this study identifies several challenges that necessitate resolution in the existing zero trust framework.

### 5.1. Zero Trust Theory

#### 5.1.1. Establish Initial Trust

Achieving the initial trust in zero trust is a problem that needs to be solved. Currently, most researchers of zero trust assume that the trust granter owns information about the requesters and resource owners that ensures the validation of authentication and authorization. However, it should be noted that the trust granter cannot recognize the parties in the transaction without information about them [55]. Hence, the mechanism for establishing the initial trust is of paramount importance in the context of zero trust. There are two feasible approaches based on existing trust mechanisms. First is the adherence to the original trust assumptions while augmenting the monitoring of transactional and environmental information between both parties. This entails utilizing multi-source data to adjust the trust relationship dynamically, thereby creating a dynamic trust adjustment mechanism. Second is the modification of the scope of privileges during the initial trust phase by granting users certain non-critical privileges and resources. This allows for the acquisition of unique information such as identity and behavioral patterns as a basis for establishing trust.

#### 5.1.2. Dynamic Trust Mechanism

In the context of zero trust, trust is established and assessed through the processes of authentication and authorization conducted by the trust granter. These processes adhere to a binary categorization: trust and untrust. However, in alignment with the principles underlying trust, there is a need for a dynamic hierarchy of trust that takes contextual factors into account. Therefore, future research can explore two key aspects: determining the weightage of codes of conduct in different contexts, and leveraging past behavioral records. Social research has demonstrated that the weight attributed to codes of conduct diminishes as trusted transactions increase [17]. Given that trust assurance in zero trust often relies on individual transactions, it becomes necessary to incorporate the evaluation of previous transaction records in addition to predefined policies [56]. When considering historical behavioral records, it is imperative to eliminate the influence of irrelevant records on trust within the zero trust framework, and instead utilize relevant records to evaluate the current transaction.

#### 5.1.3. Insider Threat of Zero Trust

Furthermore, an in-depth exploration of insider threats plays a critical role in advancing the understanding of zero trust. Therefore, defining the concept of the insider threat within the context of zero trust is a significant concern in this field. Drawing insights from real-life instances of insider incidents, researchers have employed the following definition for insider threat: “an insider threat is characterized by an individual with privileged access who either misuses these privileges or inadvertently facilitates their misuse” [57]. Firstly, it is worth highlighting that the term “privileged individual” encompasses both malicious and non-malicious insiders. Secondly, the insider threat arises as a result of either intentional misuse or inadvertent leakage of privileged access. Intentional misuse refers to the deliberate exploitation of privileges with awareness of their authorized nature. Inadvertent leakage, on the other hand, refers to the exploitation of unauthorized privileges due to vulnerabilities in the security system. Lastly, the consequences of an insider threat typically manifest as observable alterations of specific information, such as system or machine status and metrics. Consequently, the detection of insider threats within a zero trust framework can be approached from three distinct dimensions: the insiders themselves, i.e., the individuals involved; the behavioral patterns generating the threat, i.e., the operational actions executed by these individuals on the system or machine; and the manifestation of the threat outcome, i.e., the response of the system or machine to these behavioral patterns.

#### 5.1.4. Entropy of Zero Trust

Compared to traditional security theories, zero trust theory requires more granular identity authentication and access control, which necessitates the analysis of larger-scale and more detailed data. However, these vast amounts of information are highly heterogeneous, containing numerous redundant, erroneous, and irrelevant noise. As a core concept in information theory, entropy theory has been extensively studied by researchers and offers distinct advantages in evaluating and quantifying uncertainty and information heterogeneity. Consequently, researchers in the field of zero trust can employ entropy theory as a quantitative assessment tool for various aspects such as identity authentication, access control, and threat intelligence. In the context of zero trust identity authentication, entropy theory can be applied to evaluate and measure the trustworthiness of different identity authentication mechanisms. By calculating the entropy values of various authentication factors, it becomes possible to determine the diversity and heterogeneity of the authentication information, ultimately selecting more reliable and secure information for participation in the authentication process. For zero trust access control, entropy theory can be employed to evaluate and compare the entropy values of different permission allocation schemes, enabling the selection of optimal permission allocation strategies and facilitating more fine-grained access control. Moreover, entropy theory can be utilized in zero trust threat intelligence to analyse and quantify the degree of heterogeneity in network traffic data. By calculating the entropy value of network traffic, a benchmark model can be established for detecting abnormal traffic, thereby assisting in the identification of anomalous behaviors and intrusion events.

### 5.2. Zero Trust Application

#### 5.2.1. Zero Trust in Cloud

In cloud environments, the challenges faced by zero trust solutions go beyond deployment difficulties and involve effectively utilizing or adapting to the specific characteristics of the cloud. One challenge is the complex network topology inherent in the cloud. Cloud environments typically consist of multiple cloud service providers and deployment regions, requiring the identification and verification of components within the environment and ensuring a secure communication between them. To address this challenge, zero trust solutions could focus on dynamic network segmentation and rapid fine-grained mechanisms, dynamically constructing business networks and synchronously adjusting access policies based on the required services. Another challenge arises from the diversity of information generated in the cloud. The multitude and frequency of business transactions in the cloud allows for the implementation of various identity verification mechanisms. This provides multiple options for zero trust solutions to choose from, but also presents the challenge of determining which authentication mechanism is the most appropriate. The presence of a large-scale user and service population in the cloud constitutes another challenge, resulting in the need for managing identities and permissions at scale. Ensuring the efficient execution of cloud services, zero trust solutions need to consider which management mechanisms can swiftly locate the required identity information and control policies, thus reducing the latency impact caused by frequent identity authentication and permission control.

#### 5.2.2. Zero Trust in IoT

In the Internet of Things, zero trust solutions face significant challenges in the areas of the network environment, computing resources, privacy security, and communication efficiency. Firstly, the IoT serves as an underlying network environment that supports the free entry and coexistence of devices, leading to a wide variety of devices from different manufacturers and types. Furthermore, these devices may operate on diverse communication protocols and operating systems. The continual identity authentication and fine-grained access control required by zero trust solutions are greatly constrained under these conditions. As a result, it is necessary to establish universal protocol standards and access control policies to accommodate these different types of devices while considering the scalability requirements of the IoT. Additionally, the IoT exhibits significant variations in resource availability and computing capabilities among devices, particularly with a substantial number of resource-constrained devices. Hence, it is crucial to consider how to achieve secure authentication and access control on resource-limited devices when designing zero trust solutions. Moreover, although IoT data are primarily collected or generated by network devices, they encompass a vast amount of sensitive information such as user privacy and trade secrets. If this information were to be compromised, attackers would quickly identify target devices and conduct threatening actions. Therefore, measures must be implemented during the implementation of zero trust in the IoT to ensure data privacy and security, preventing unauthorized access and data leakage. Lastly, the real-time requirements of the IoT pose significant challenges to zero trust solutions. Despite the limited computing resources in the IoT, many applications demand real-time processing and responsiveness. Directly applying existing zero trust mechanisms for identity authentication and access control would inevitably impact IoT performance. Consequently, zero trust solutions in the IoT must carefully consider the influence of zero trust on communication efficiency, ensuring that the design and implementation of these solutions meet real-time requirements without significant performance impacts.

## 6. Conclusions

In response to the escalating internal threat incidents, the concept of a trusted internal network within the scope of a traditional network perimeter can no longer be regarded as secure. Consequently, the zero trust paradigm has emerged, aiming to eradicate the reliance on implicit trust in networks and systems. This paper conducted a survey on the theory and application of zero trust security, organizing and summarizing the fundamental theories and architectural frameworks, as well as the application of zero trust in cloud computing and the IoT. Diverging from other survey articles that primarily focus on the detailed implementation aspects of zero trust, this paper attempts to analyze the essence of zero trust from a more fundamental perspective, i.e., what trust means within the context of zero trust. The review of the literature reveals that zero trust fundamentally entails a continuous suspicion of the trustworthiness of implicit trust. In other words, the formerly implicit trust can no longer be deemed reliable, necessitating the transformation of implicit trust into explicit trust through technological means or verification mechanisms. Consequently, this paper proposes a novel concept, namely the “trustbase”, serving as the foundation of explicit and implicit trust. When both explicit and implicit trust fail simultaneously, the trustbase can serve as a substitute for trust to fulfill the minimal security requirements. Furthermore, this paper examines the future research trends and challenges of zero trust from the perspective of trust. The results indicate that both the theoretical concepts and technical aspects of zero trust revolve around the fundamental questions of how to establish trust and how to verify trust. Thus, this paper presents several research aspects within zero trust security and provides reference research ideas and methods, aiming to assist readers in identifying intriguing and challenging research topics. The intention is for the content of this paper to facilitate beginners in zero trust research to attain the essential knowledge required for studying zero trust, as well as provide dependable references for conducting related research.

## Figures and Tables

**Figure 1 entropy-25-01595-f001:**
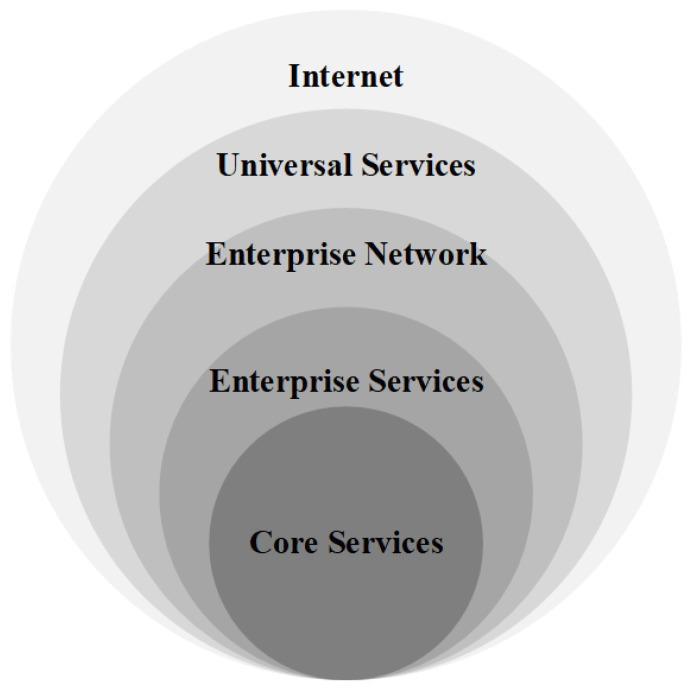
A secure perimeter network for enterprises.

**Figure 2 entropy-25-01595-f002:**
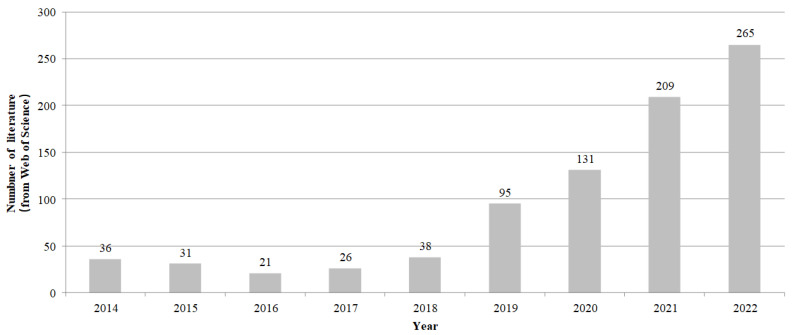
Number of zero trust literature from 2014 to 2022.

**Figure 3 entropy-25-01595-f003:**
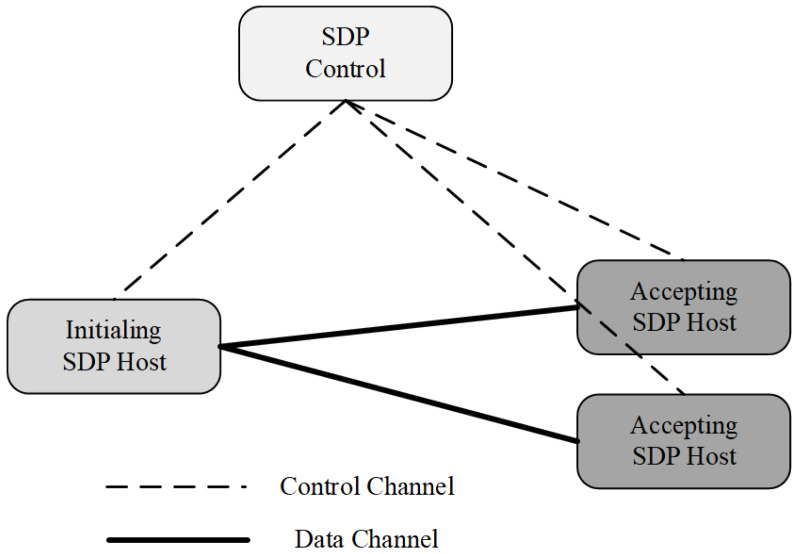
The architecture of the SDP.

**Figure 4 entropy-25-01595-f004:**
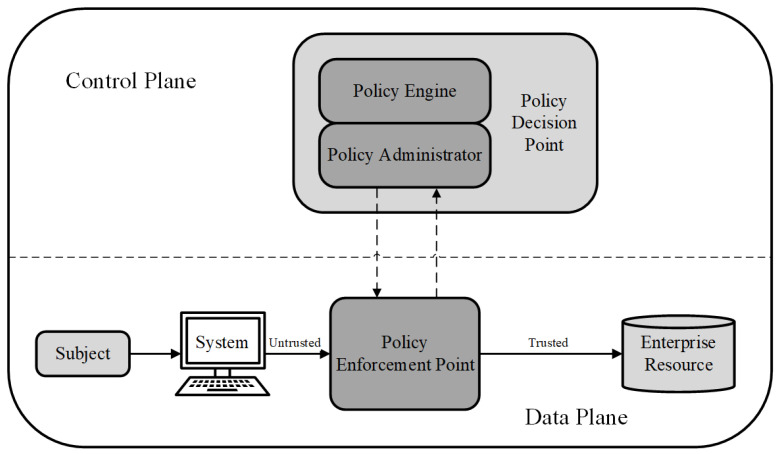
The logical components of ZTA.

**Figure 5 entropy-25-01595-f005:**
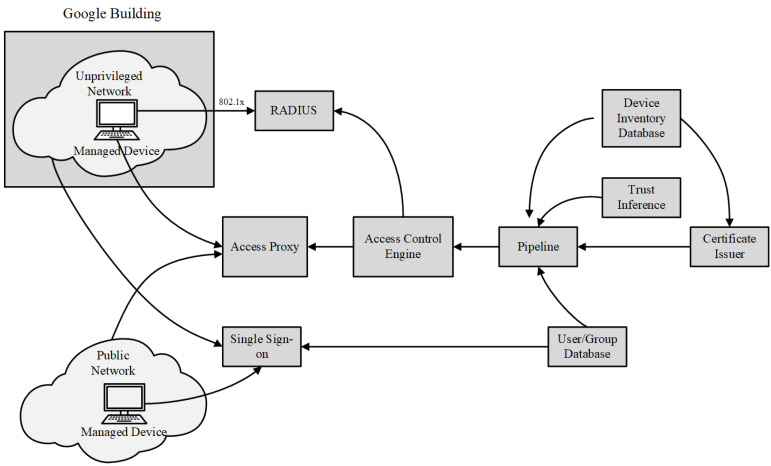
BeyondCorp components and access flow.

**Figure 6 entropy-25-01595-f006:**
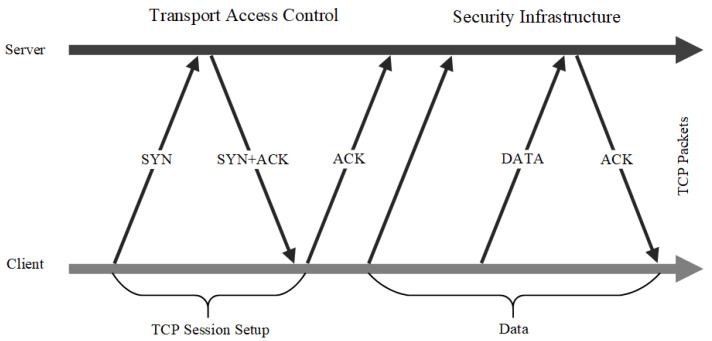
TAC approach of Transport Layer.

**Figure 7 entropy-25-01595-f007:**
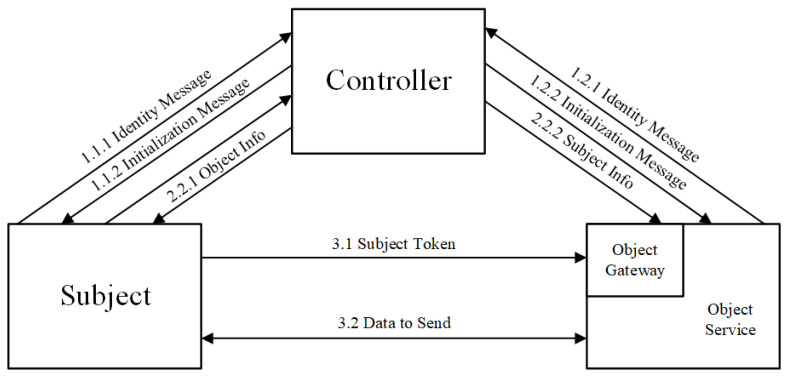
Basic zero trust architecture.

**Figure 8 entropy-25-01595-f008:**
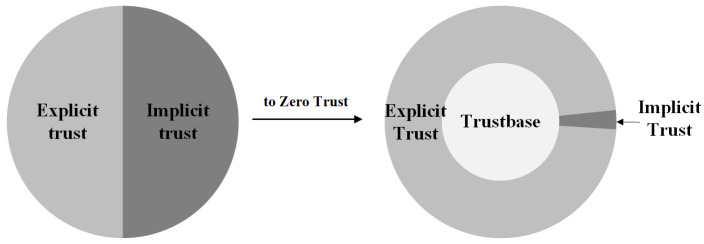
The change in trust composition.

**Table 1 entropy-25-01595-t001:** Overview of representative literature on zero trust theory.

Author(s)	Year	Title	Publication Type	View
Kindervag, J. [3]	2010	No more chewy centers: Introducing the zero trust model of information security	Institution Research Report	Trust is a vulnerability and, like all vulnerabilities, should be eliminated; zero trust should consider the flaws of security perimeters facing internal threats and the risks arising from implicit trust.
Cloud Security Alliance [8]	2013	Software Defined Perimeter	Institution Publication	A new approach is needed that enables application owners to protect infrastructure in a public or private cloud, a server in a data center, or even inside an application server.
Ward, R.; Beyer, B. [5]	2014	Beyondcorp: A new approach to enterprise security	Institution Research Report	The perimeter is no longer just the physical location of the enterprise, and what lies inside the perimeter is no longer a blessed and safe place to host personal computing devices and enterprise applications; fat-client applications that use proprietary protocols to talk to servers will be a challenge for moving to BeyondCorp.
Osborn, B.; McWilliams, J.; Beyer, B.; Saltonstall, M. [6]	2016	Beyondcorp: Design to deployment at Google	Institution Research Report	Correct trust assessment requires ensuring data quality and relevance; the sparsity of data sets will hinder the use of high-productivity applications, and it is necessary to monitor and verify whether these data can produce expected results when used for trust assessment; the unique implementation of obtaining information from multiple data sources requires simplifying the data dissemination method to control latency; fundamental changes to the security infrastructure can potentially adversely affect the productivity of the entire company’s workforce; various fail-safes need to be deployed to reduce the impact of catastrophic failures.
Kindervag, J. [21]	2016	No more chewy centers: The zero trust model of information security	Institution Research Report	Perimeter-based network security models fail to protect against today’s threats; eliminate chewy centers with the zero trust model; zero trust is not a one-time project.
Escobedo, V.; Beyer, B.; Saltonstall, M.; Zyzniewski, F. [7]	2017	BeyondCorp: the user experience	Institution Research Report	The deployment of BeyondCorp needs to take into account the user’s learning cost, user experience and workflow, which is also what other zero trust solutions need to focus on.
Eidle, D.; Ni, S.Y.; DeCusatis, C.; Sager, A. [25]	2017	Autonomic security for zero trust networks	Conference Paper	Automated systems could improve both efficiency and response time to immediate threats; increased automation of cyber-defense leveraging OODA, and autonomic principles holds potential for defending cloud and enterprise data center networks.
ACT-IAC Zero trust Project Team [19]	2019	Zero trust Cybersecurity Current Trends	Institution Research Report	Zero trust is not a technology in and of itself but a shift in the design approach for cybersecurity; zero trust can augment and compliment other cybersecurity tools and practices rather than replacing them.
Moubayed, A.; Refaey, A.; Shami, A. [26]	2019	Software-Defined Perimeter (SDP): State of the Art Secure Solution for Modern Networks	Journal Paper	SDP faces the challenges of security, privacy and availability; exploring how to integrate SDP with other paradigms such as SDN and NFV is essential.
Kumar, P.; Moubayed, A.; Refaey, A.; Shami, A.; Koilpillai, J. [27]	2019	Performance analysis of sdp for secure internal enterprises	Conference Paper	SDP suffers from long connection setup time but it can provide robustness to the network under threats; SDP provides protection against a wide range of attack, it cannot guarantee complete protection.
Rose, S.; Borchert, O.; Mitchell, S.; Connelly, S. [9]	2020	Zero trust architecture	Institution Publication	Zero is not a single architecture but a set of guiding principles for workflow, system design and operations; Organizations should seek to incrementally implement zero trust principles, process changes, and technology solutions; Organizations need to implement comprehensive information security and resiliency practices for zero trust to be effective.
Campbell, M. [24]	2020	Beyond zero trust: Trust is a vulnerability	Journal Editorial	The attack surface is never static, never localized, and never impregnable; zero trust solutions will mature and become the security strategy standard as they grow more automated, smart, and extended.
Singh, J.; Refaey, A.; Shami, A. [28]	2020	Multilevel security framework for nfv based on software defined perimeter	Journal Paper	There is a need for an additional security framework to improve the NFV security solutions; the install deployment of SDP has problems such as difficulty in certificate distribution and complicated installation process; the potential for attacks originating from inside a secured network is an open challenge which NFV-SDP could mitigate, in theory.
NSA Cybersecurity Requirements Center [22]	2021	Embracing a Zero trust Security Model	Institution Research Report	The scalability of the capabilities is essential for applying zero trust; implementing Zero Trust should not be undertaken lightly and will require significant resources and persistence to achieve.
Garbis, J.; Chapman, J.W. [23]	2021	What Is Zero Trust?	Book Chapter	A Zero Trust system is an integrated security platform that uses contextual information from identity, security, and IT Infrastructure; and risk and analytics tools to inform and enable the dynamic enforcement of security policies uniformly across the enterprise.
Syed, N.F.; Shah, S.W.; Shaghaghi, A.; Anwar, A.; Baig, Z.; Doss, R. [29]	2022	Zero trust architecture (zta): A comprehensive survey	Journal Paper	A lightweight and scalable continuous authentication mechanism is essential to achieve trust for resources; a fine-grained contextual access control scheme is needed to adapt to different network environments; the goal of ZTA is to protect data, and encryption is an important requirement to achieve zero trust; ZTA requires micro-segmentation to prevent attackers from lateral movement, and the single point of failure issues need to be addressed; an effective feedback system is needed to provide ZTA with threat intelligence and security situational awareness; ZTA requires reliable trust assessment capabilities to implement dynamic access control and could use ML to provide automatic learning capabilities; the fuzziness and heterogeneity of data require a more variable trust mechanism.
Leahy, D.; Thorpe, C. [37]	2022	Zero Trust Container Architecture (ZTCA): A Framework for Applying Zero Trust Principals to Docker Containers	Conference Paper	The security issues of Docker deployment can be solved based on whether the components deployed by Docker belong to trust zones rather than focusing on specific attacks; the implicit trust that Docker users place on the Docker engine is a recipe for security issues, and this is exactly what ZTA can alleviate.

**Table 2 entropy-25-01595-t002:** Overview of representative literature on zero trust application.

Author(s)	Year	Title	Publication Type	View
DeCusatis, C.; Liengtiraphan, P.; Sager, A.; Pinelli, M. [39]	2016	Implementing zero trust cloud networks with transport access control and first packet authentication	Conference Paper	Traditional VLANs and similar network segmentation technologies do not provide sufficient network security; penetration testing is required to identify and mitigate any additional vulnerabilities of the scheme.
Samaniego, M.; Deters, R. [46]	2018	Zero trust hierarchical management in IoT	Conference Paper	Neither the infrastructure in IoT nor the transactions performed by the infrastructure can be trusted; the resource-constrained environment of IoT makes zero trust solutions using a central trust verification authority difficult to implement.
Zaheer, Z.; Chang, H.; Mukherjee, S.; Van der Merwe, J. [40]	2019	eZTrust: Network-independent zero trust perimeterization for microservices	Conference Paper	Traditional perimeterization approaches do not fare well in highly dynamic microservices environments; eZTrust could expand in tag granularity, tag anonymization, smart NIC offload and platform compatibility.
Dhar, S.; Bose, I. [47]	2021	Securing IoT devices using zero trust and blockchain	Journal Paper	The implementation of standard security mechanisms, such as access control, session management, and cryptography mechanism, is difficult for IoT networks; ways need to be found to measure the reliability of the proposed framework in real-world IoT networks.
Zhao, S.; Li, S.; Li, F.; Zhang, W.; Iqbal, M. [48]	2021	Blockchain-enabled user authentication in zero trust internet of things	Conference Paper	The heterogeneity and security of IoT require each device to authenticate before being granted access, ensuring serverless, passwordless, self-sovereign security.
Chen, Z.; Yan, L.; Lü, Z.; Zhang, Y.; Guo, Y.; Liu, W.; Xuan, J. [53]	2021	Research on zero trust security protection technology of power IoT based on blockchain	Conference Paper	Traditional network security technology that relies on the central organization cannot meet the high security requirements of the development of the new energy Internet business; the power terminal has a wide deployment range, uncontrolled on-site environment, and complex vulnerability risk handling that will provide attackers with opportunities to threat energy network.
Zhang, X.; Chen, L.; Fan, J.; Wang, X.; Wang, Q. [54]	2021	Power IoT security protection architecture based on zero trust framework	Conference Paper	The gradual implementation of the power IoT will bring new demands such as massive access, heterogeneous authentication, and frequent interactions, and existing security protection methods are not enough to cope with them; in the construction of the power IoT, in addition to considering the identity management of people, it is also necessary to authenticate the identity of devices, applications, and services.
Liu, S.; Zhuang, Y.; Huang, L.; Zhou, X. [41]	2022	Exploiting lsb self-quantization for plaintext-related image encryption in the zero trust cloud	Journal Paper	Existing plaintext-related image encryption schemes cannot meet the requirements of ZT in both reducing implementation overhead and resisting multiple attacks.
Zolotukhin, M.; Hämäläinen, T.; Kotilainen, P. [42]	2022	Intelligent Solutions for Attack Mitigation in Zero Trust Environments	Journal Paper	The requirement for dynamic updates of access policies in IoT requires that deployed zero trust models dynamically adjust security policies to reduce the ongoing attack surface and minimize the risk of subsequent attacks; traffic from real devices and applications is needed to validate the effectiveness and scalability of the framework.
Sarkar, S.; Choudhary, G.; Shandilya, S.K.; Hussain, A.; Kim, H. [43]	2022	Security of zero trust networks in cloud computing: A comparative review	Journal Paper	Zero Trust can also be combined and used with other novel technologies such as blockchain and the IoT; zero trust under 5G/6G networks can use AI to quickly prevent malicious requests and network performance degradation to serve mission-critical areas such as healthcare, military, and autonomous driving; zero trust can serve national security needs as a technology to prevent adversaries from entering military networks; using zero trust in containerized software, microservices, and sustainable cloud systems requires greater focus on performance, security, reliability, and sustainability.
Alevizos, L.; Ta, V.T.; Hashem Eiza, M. [49]	2022	Augmenting zero trust architecture to endpoints using blockchain: A state-of-the-art review	Journal Paper	A compromised endpoint’s authenticated and authorized session can perform limited activities, becoming ZTA’s Achilles’ heel; using blockchain-based intrusion detection and authentication in ZTA requires full consideration of issues such as performance, computational overhead, and appropriate blockchain implementation.
Palmo, Y.; Tanimoto, S.; Sato, H.; Kanai, A. [50]	2022	Optimal Federation Method for Embedding Internet of Things in Software-Defined Perimeter	Journal Paper	When embedding IoT devices into SDP, it is imperative to ensure the reliability of the IoT device.
Valero, J.M.J.; Sánchez, P.M.S.; Lekidis, A.; Hidalgo, J.F.; Pérez, M.G.; Siddiqui, M.S.; Celdrán, A.H.; Pérez, G.M. [51]	2022	Design of a Security and Trust Framework for 5G Multi-domain Scenarios	Journal Paper	5G networks require security and trust mechanisms covering multi-domain scenarios to achieve complete isolation of network slices, thereby preventing unauthorized and malicious entities from accessing 5G infrastructure; for 5G networks, the zero trust approach can not only guarantee data and services intra-and inter-domain protection, but also all enterprise resources and subjects, thereby enclosing the security and trust threat landscape.
Li, S.; Iqbal, M.; Saxena, N. [52]	2022	Future industry internet of things with zero trust security	Journal Paper	Developing security policies and hybrid policy definitions to be followed on 5G networks is very important for the application of zero trust model on 5G; The zero trust model’s continuous monitoring and analysis of each device hinders latency, which in turn affects its use in IoT.

## Data Availability

No new data were created or analyzed in this study. Data sharing is not applicable to this article.

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
