# Peer review of "Theory and Application of Zero Trust Security: A Brief Survey"

_entropy, 2023, doi:10.3390/e25121595_

Round 1

Reviewer 1 Report

Comments and Suggestions for Authors

In this paper, the authors present descriptively the concepts inherent to network security by applying the theory and application of zero trust security. Network security is a relevant and interesting topic. However, the current version of the paper does not present significant results from scientific research or contribute to the body of knowledge. The paper, in its current version, is similar to a review. However, it does not present a systematic review research strategy. In its current form, the paper's contribution to Entropy's knowledge area is unclear. The authors do not present practical applications or comparative results demonstrating advantages and disadvantages with other network security methodologies. Considering the number of corrections and inclusions of new information in the paper to be considered for publication, I recommend that the paper be rejected and that the authors rewrite a paper in the form of a review and adopt a systematic method of obtaining the works to be cited.

I recommend the following corrections:

1- Adapt the title of the paper for a review work. As an example, we can cite “A Short Note on Theory and Application of Zero-trust Security”.

2- A table identifying the types of research and references of authors who published on the topic must be included;

3- A table with the positive, negative points and future perspectives correlating the cited authors must be included in the paper;

4- A table with the temporal growth of published works must also be included in the body of the paper;

5- A systematic research tool must be considered and cited in the paper to indicate the methodology adopted in the search for published works;

6- The conclusions must be expanded considering the new results included in the paper.

Author Response

Dear Reviewer,

Thanks very much for taking your time to review this manuscript. We really appreciate all your comments and suggestions! These suggestions have enabled us to improve our work. Based on the instructions provided in your letter, we have uploaded a revision of the manuscript in the system.

Appended to this letter is our point-by-point response to the comments raised by the reviewer. The comments are reproduced and our responses are given directly afterward in a different color (blue).

We would like also to thank you for allowing us to resubmit a revised copy of the manuscript.

We hope that the revised manuscript is accepted for publication in the Entropy.

Sincerely,

Corresponding author: Gang Liu

School of Computer Science and Technology, Xidian University

266 Xinglong Section of Xifeng Road, Xi’an, Shaanxi 710126, P. R. China

Reviewer 2 Report

Comments and Suggestions for Authors

The paper addresses a very interesting and novel topic such as zero-trust security. This is a survey on theory and technical applications in Cloud and IoT environments. This also includes a discussion on trust in cybersecurity and zero trust. Nevertheless, some improvements could be included:

- A comparison between proposals for applications in both cloud and IoT should be more shown. In addittion, different aspects involved in a zero trust architecture are not analysed. This could have a more practical approach.

- The analysis and research trends or challenges should be also explained respect to application scenarios.

- The format of references should be reviewed.

Comments on the Quality of English Language

The quality of english language is correct.

Author Response

(The authors gave the same response as above.)

Reviewer 3 Report

Comments and Suggestions for Authors

This article presents a survey on the main aspects of zero-trust security. The research area is novel and challenging. The paper is easy to follow. However, there are some sections that should be better presented and highlight more the contribution of the survey. In order to improve the quality of their work, the author should address the following comments.

1. The last paragraph of Introduction must clearly present the contributions of the survey and its novelty against other survey on zero-trust.

2. Section 5 should better present the research challenges and future research directions. A breakdown to subsection will be helpful towards this direction.

3. Section 3.2: Correct the references on this section. the numbering is wrong.

Comments on the Quality of English Language

-

Author Response

(The authors gave the same response as above.)

Round 2

Reviewer 1 Report

Comments and Suggestions for Authors

The authors carried out an extensive review of the paper, submitting a new version with the requested suggestions and corrections. Considering the extensive correction made and the new content presented, I consider that the current version is capable of being published.

Reviewer 3 Report

Comments and Suggestions for Authors

All of my previous comments are properly addressed.